# Improvement of Temperature and Optical Power of an LED by Using Microfluidic Circulating System of Graphene Solution

**DOI:** 10.3390/nano11071719

**Published:** 2021-06-29

**Authors:** Yung-Chiang Chung, Han-Hsuan Chung, Shih-Hao Lin

**Affiliations:** Department of Mechanical Engineering, Ming Chi University of Technology, New Taipei 24301, Taiwan; gary850311@gmail.com (H.-H.C.); M07118021@mail2.mcut.edu.tw (S.-H.L.)

**Keywords:** liquid conductor, graphene solution, circulating system, microfluidic channel, temperature, optical power

## Abstract

Electric devices have evolved to become smaller, more multifunctional, and increasingly integrated. When the total volume of a device is reduced, insufficient heat dissipation may result in device failure. A microfluidic channel with a graphene solution may replace solid conductors for simultaneously supplying energy and dissipating heat in a light emitting diode (LED). In this study, an automated recycling system using a graphene solution was designed that reduces the necessity of manual operation. The optical power and temperature of an LED using this system was measured for an extended period and compared with the performance of a solid conductor. The temperature difference of the LED bottom using the solid and liquid conductors reached 25 °C. The optical power of the LED using the liquid conductor was higher than that of the solid conductor after 120 min of LED operation. When the flow rate was increased, the temperature difference of the LED bottom between initial and 480 min was lower, and the optical power of the LED was higher. This result was attributable to the higher temperature of the LED with the solid conductor. Moreover, the optical/electric power transfer rate of the liquid conductor was higher than that of the solid conductor after 120 min of LED operation, and the difference increased over time.

## 1. Introduction

Heat dissipation and electrical conduction are critical considerations in the operation of an integrated circuit device. Solid metal conductors are typically used for conducting electricity. Several heat dissipation methods, such as the use of microfluidic devices, are employed in electric systems. For example, researchers have considered using nanofluids for cooling electric devices [1,2]. Studies have proposed a thermal contact liquid cooling system [3] as well as a technique for cooling photovoltaic cell systems through the use of rotating magnetic fields and ferrofluids [4]. A related study used a system combining liquid cooling and composite phase change material cooling to dissipate the heat generated in a battery [5]. Liquid cooling systems have also been applied in central processing units and laptops [6,7]. Research has revealed various cooling methods for electric devices [8,9,10]. Some studies have also employed the recycling of various fluids, such as ammonia [11], CO_2_ [12], and supercritical water [13] for cooling. Additionally, researchers have introduced a one-section and two-stepwise microchannel for cooling [14] and a recirculating cooling water system to reduce energy consumption [15]. Another study considered the cooling effect of dielectric liquid [16], and yet another reported that a fracture in the neck of the bond between the solder joint and gold wire after 20 thermal shocks would result in metal conductors failing at high temperatures [17].

The dissipation of heat in light emitting diodes (LEDs) has been studied extensively. For example, researchers have investigated the temperature distribution [18] and liquid cooling systems [19,20] of an LED array. Furthermore, composite coatings composed of cupric oxide [21] or various heat pipes and heat sinks [22,23,24,25,26,27,28] have been employed to enhance the heat dissipation of LEDs. Other relevant studies have used a dielectric layer with an aluminum nitride insulation plate [29] and a dual synthetic jet actuator for heat dissipation in LEDs [30]. A cooling system with ferrofluid was used in a high-power LED; the results were compared with those of systems employing air and water working fluid, and the effect of ferrofluid was the best [31]. A thermoelectric cooler integrated with a microchannel heat sink was used to control LED temperature [32]. Another study proposed the energy recycling and self-sufficient application of an LED integrated with a thermoelectric generator module and electrical fan [33]. Moreover, graphene has been employed as a novel material in several investigations [34,35]. Some researchers have considered the thermophysical properties and forced convective heat transfer performance of graphene [36,37] and reviewed the applications of graphene [38,39]. Researchers have suggested that the conductivity of water can be improved through the addition of mono and hybrid nano-additives containing graphene and silica [40] and have developed a method that integrates graphene nanocapillaries into a micro heat pipe for enhanced LED cooling [41]. Graphene solutions have been widely applied to increase heat transfer efficiency [42,43,44,45,46,47], and a higher graphene solution concentration has been demonstrated to result in greater heat transfer [48,49]. Moreover, some researchers have studied thermal conductivity and electrical conductivity of graphene nanoplatelets [50,51]. The aforementioned investigations have concentrated on either dissipating heat or supplying energy but have not considered the combination of energy supply and heat dissipation. Furthermore, numerous studies have examined the energy conversion of artificial light [52,53,54,55] and white light LEDs [56,57,58]. Evidence suggests that the power conversion efficiencies of solar cells and LEDs are lower than 53.6% [59,60] and that the efficiency of LEDs is 42% at 30 °C, dropping to 30% at 50 °C [61]. However, the aforementioned studies focused predominantly on energy conversion from optical power to electric power; energy conversion from electric power to optical power has seldom been discussed. The stability and reliability of an electric apparatus may decrease by 10% when the temperature is increased by 2 °C [62]. In another investigation, a graphene solution was used as a liquid conductor for dissipating heat and transferring energy; the heat dissipation efficiency was excellent, but the optical power of an LED with a liquid conductor was lower than that of an LED with a solid conductor during the first 6 min, and the graphene solution was recycled manually [63]. According to the aforementioned study, the temperature of the LED using the liquid conductor was much lower than that of the LED using the solid conductor. However, the optical power of LEDs using liquid and solid conductors was not compared over longer periods. Therefore, to supply energy and dissipate heat of an LED during longer experiments, the graphene solution can be automatically recycled. Such a system can simultaneously dissipate heat and supply energy as well as improve the LED’s optical power. The optical power of the LED using the liquid conductor may be higher than that of the LED using the solid conductor over longer periods. Therefore, in this study, an automated liquid conductor circulating system was developed and the energy supply and heat dissipation of an LED was studied over an extended period.

## 2. Materials and Methods

### 2.1. Principle of Operation

A liquid conductor that can dissipate heat and conduct electricity simultaneously may effectively reduce the temperature of electric products and enhance the photoelectricity transfer efficiency. Related research has indicated that the temperature of an LED using a liquid conductor is much lower than that of an LED using a solid conductor. Thus, the effect of the temperature of the LED on its optical power, especially during extended periods, merits further study.

### 2.2. Chip Design and Fabrication

Microfluidic channels were fabricated using a microelectromechanical process, as displayed in Figure 1a,b. The widths of the inlet and outlet of the channel were 0.5 mm, and the length of the channel was 20 mm; the distance between the channels was 0.5 and 1 mm, respectively. As displayed in Figure 2a, first, the wafer was cleaned using acetone and deionized water. Second, the periphery of wafer was coated with Teflon and baked; the periphery of the wafer was hydrophobic to enhance the photoresist coating. The silicon wafer was coated with SU8 photoresist with a thickness of approximately 500 μm at 200 rpm. Subsequently, the wafer was soft baked and the solvent in the photoresist was removed. The wafer was exposed to define the pattern, and it was baked after exposure to enhance the linking process of the photoresist. After the wafer was developed, the developer removed the undefined region. Subsequently, the polydimethylsiloxane (PDMS) substrate microfluidic channel chip was fabricated. As illustrated in Figure 2b, the Teflon was coated on the wafer and the PDMS was poured into the wafer; the wafer was hydrophobic to facilitate the fabrication of the PDMS substrate. The wafer was then vacuumed to remove the bubbles in the PDMS, and the wafer was heated to solidify the PDMS. Once the wafer had cooled, the PDMS could be separated from the wafer, and the PDMS substrate microfluidic channel chip was complete.

The designated area for a microscope slide (75 × 25 × 1 mm^3^) was coated with silver adhesive (OP-901, Double O Technology, Taiwan), as shown in Figure 3a, which was connected to the microfluidic channel and the LED. The PDMS substrate microfluidic channel chip and the microscope slide coated with silver adhesive were bonded together to form the chip. The LED (emission color: white; TY-HNW2-3, TaoYuan Electron Limited, Taiwan) was placed at the center of the chip. The silver adhesive was coated on the electrodes of the LED, connecting it with the silver adhesive of the chip. The size of the liquid recycling reservoir was 75 × 25 × 13 mm^3^. The area between the needle and the microfluidic channel was sealed with ultraviolet glue to ensure that the liquid would not leak. Finally, the chip, the liquid or solid conductor, the liquid recycling reservoir, and the LED were integrated to form an LED with a liquid circulating system, shown in Figure 3b.

### 2.3. Sample Preparation and Experimental Setup

A graphene solution (Golden Innovation Business Co., Ltd., New Taipei, Taiwan) was used as the liquid conductor in this study. The size of the graphene particles in the solution was 100 nm, the solution was deionized water, and the concentration was 500 ppm. The needle was placed into the microfluidic channel. The graphene solution was fully mixed using an acoustic vibrator, and the solution was injected into the needle and microfluidic channel. The flow rate was regulated using a piezoelectric micropump (CurieJet, PS51I, Microjet Technology, Hsinchu, Taiwan); the flow rate ranged from 0.01 to 10 mL/min. A solid conducting nickel-plated steel wire (diameter: 1 mm) was used as the solid conductor. The power supply system (LPS305, Motech, Tainan, Taiwan) could produce an output voltage of ± 30 V. The LED system included the power supply, a syringe pump, the chip, and the graphene solution (or solid conductor). The temperature was measured using resistance temperature detectors (RTDs; PT 100 series, OMEGA Engineering Inc., Norwalk, Connecticut, USA), and the optical power of the LED was measured using an optical power meter (Gentec Electro-Optique Inc., Quebec, Quebec, Canada). The integrated system (Figure 4) included the LED system, power supply, syringe pump, temperature data receiver, and optical power sensor. In this study, the temperature and optical power of the LED were measured over an extended period, and the effect of the temperature on the optical power was examined.

## 3. Results and Discussion

In this study, liquid (graphene solution) and solid (nickel-plated steel wire with a diameter of 1 mm, length of 30 mm) conductors were used. The temperatures at the bottom of the LED chip and the microfluidic channel were measured. Moreover, the optical power of the LED under various conditions was measured at a distance of 3 mm from the LED. The uncertainty analysis results of various parameters are listed in Table 1. Experiments were performed five times under each set of experimental conditions, and the average error was less than 20%. The temperatures at the center and four corners were measured, and the temperature variation was lower than 2 °C. Because the temperature at the center of the chip bottom was the highest, it was selected to represent the temperature of the LED chip (Figure 5).

### 3.1. Energy Supply

The actual voltage of the LED with the graphene solution conductor differed from the voltage of the power supply. The actual voltages of the LED with the graphene solution were measured at various power supply voltages (Figure 6). The results indicated that the voltage of the graphene solution–conductor LED was 2.8 V when the voltage of the power supply was 5.5 V. The highest operational voltage of the LED was 3 V. Therefore, the actual voltage of the LED with the solid and liquid conductors was selected as 2.8 V in this study.

### 3.2. Temperature Variation Using Different Conductors

The measured temperatures of the LEDs with solid and liquid conductors are listed in Figure 7. The temperatures of the LED bottom and the microfluidic channel bottom increased over time. For the solid conductor, the temperatures of the LED bottom increased to 58 °C at 50 min, and the temperatures increased gradually to 65.5 °C at 480 min. The temperatures of the microfluidic channel bottom increased to 51.5 °C at 50 min and continued to increase gradually to 59.4 °C at 480 min. The temperature difference between the LED bottom and the microfluidic channel bottom was 6–7 °C. The measured temperatures of the liquid conductors (*d* = 500 and 1000 μm) also increased over time. When *d* = 1 mm, the temperatures of the LED bottom and the microfluidic channel bottom increased over time. The temperatures of the LED bottom increased to 37.2 °C at 50 min and continued to increase gradually to 41 °C at 480 min. The temperatures of the microfluidic channel bottom increased to 28.2 °C at 50 min and continued to increase gradually to 30.5 °C at 480 min. The temperature difference between the LED bottom and the microfluidic channel bottom was 9–10.5 °C. When *d* = 0.5 mm, the temperatures of the LED bottom increased to 36.2 °C at 50 min and increased gradually to 40 °C at 480 min. The temperatures of the microfluidic channel bottom increased to 28 °C at 50 min and increased gradually to 30 °C at 480 min. The temperature difference between the LED bottom and the microfluidic channel bottom was 8.2–10 °C. The temperatures of the LED bottom and microfluidic channel bottom were lowest in the liquid conductor with *d* = 0.5 mm, but the temperature differences between these two parts did not differ considerably between the liquid conductors with *d* = 0.5 and 1 mm. Furthermore, the temperature difference of the LED bottom between the solid conductor and the liquid conductor reached 25 °C. This result emphasizes the temperature reduction effect of the LED using the liquid conductor.

The temperature differences of the LED bottom between initial and 480 min using solid conductor and graphene solution with various flow rates are shown in Figure 8. The temperature difference using the solid conductor was 35.1 °C. The temperature differences using the graphene solution were 17.2, 13.4, 11.5 and 10.1 °C at flow rates of 0, 0.05, 0.2 and 1 mL/min, respectively. When the flow rate was increased, the temperature difference was lower and the heat dissipation of the graphene solution was excellent. The heat dissipation of the graphene solution was apparently higher than that of the solid conductor. At static state, the heat dissipation of the graphene solution was still higher than that of the solid conductor.

The temperature decreases of the LED using various cooling methods are shown in Figure 9. The temperature decrease in the LED with fin only was 3 °C, that of the LED with fan only was 8 °C, and that of the LED with fin and fan could be 10 °C [22,23]. The temperature decrease in the LED with heat pipe was larger than 7 °C [24,25,26,27], that of the LED with dielectric layer could reach 20 °C [29], and it may be larger than 25 °C using a ferrofluid [31]. In this study, the temperature decrease in the LED reached 30 °C using a graphene solution. The temperature decrease in the LED in this study was equal to or larger than that using other cooling methods.

The thermal conductivity *k_s_* of the solid conductor was about 15 W/m·°C. The thermal conductivity of graphene was about 5300 W/m·°C. The thermal conductivity of the graphene solution *k*_g_ was about 20–100 W/m·°C for various concentration for 200–1000 ppm (from graphene solution company, Golden Innovation Business Co., Ltd., New Taipei, Taiwan.).

The thermal resistance of a solid conductor can be expressed as follows [64,65]:*R*_th,s_ = 1/(*k_s_*A/Δ*x*)(1)
where Δ*x* is the distance between two locations, and A is the area between two locations. Consider the convection heat transfer coefficient *h*, the thermal resistance of a graphene solution can be expressed as follows:*R*_th,g_ = 1/(*k_g_*A/Δ*x* + *h*A)(2)

The value *h* of water was 1000–35000 W/m^2^·°C for various flow rates [64,65].

Based on the comparison between *R*_th,s_ and *R*_th,g_, the thermal resistance of the graphene solution with flow rate was smaller than that of the solid conductor, and the heat transfer of the LED was improved.

### 3.3. Optical Power Variation Using Different Conductors

The optical power at various distances between the LED and the power meter is displayed in Figure 10. The optical power of the LED did not decrease noticeably when the distance was lower than 5 mm. However, it decreased markedly when the distance was greater than 10 mm. The optical power at a distance of 3 mm from the LED was selected as the representative value. The optical power of the LEDs measured using solid and liquid conductors is displayed in Figure 11. The maximum optical power of the LED with the solid conductor was 30.5 mW at 10 min, and it decreased gradually. The optical power decreased noticeably after 120 min, and that of the LED with the liquid conductor with *d* = 0.5 mm increased gradually before 30 min and stabilized at 29 mW by 200 min. The optical power of the LED with the liquid conductor with *d* = 1 mm increased gradually before 30 min and stabilized at 28.8 mW by 180 min. The optical power of the LED with the solid conductor was greater than that of the liquid conductor until 90 min. The optical power difference of the LED with liquid conductors with various *d* values was small. The temperature of the LED with the liquid conductor was much lower than that of the LED with the solid conductor after 30 min (Figure 7). The temperature differences between the LED bottoms with the liquid and solid conductors increased over time during the first 120 min and then stabilized at approximately 24 °C. The optical power of the LED with the liquid conductor was higher than that of the solid conductor after 120 min, and the optical power difference was approximately 3 mW after 150 min. This result was attributable to the higher temperature of the LED with the solid conductor.

### 3.4. Comparison of Power Transfer

The liquid pumping power (additional energy) required to pump the graphene solution is discussed in this section. The liquid pumping power, *P_l_* (W) can be expressed as follows [44]:*P_l_* = (Δ*P*) × *Q*(3)
where Δ*P* is the pressure drop (kPa), and *Q* is the flow rate (m^3^/s).

The maximum flow rate in this study was 1 mL/min (1.67 × 10^−8^ m^3^/s) and the pressure drop was approximately 50 kPa. The pumping power was estimated to be 8.3 × 10^−4^ mW. Compared with the optical power of the LED (29 mW), the pumping power was much lower. The overall operational expenditure of the device was unaffected in this study. Moreover, the volume of the graphene solution in the recycling system was 30 mL and the cost of the graphene solution was approximately USD 60. Thus, the cost of the graphene solution was not high, and the graphene solution could be automatically recycled. The cost of the graphene solution did not increase noticeably because it required no replenishment in this study. Such a system is highly convenient and avoids substantial increases in the total device cost. Therefore, a graphene-solution microfluidic channel is a favorable conductor for use in LEDs.

The optical power of the LED using the solid conductor and graphene solution with various flow rates at 480 min are shown in Figure 12. The optical power of the LED using the solid conductor was 24.5 mW. The optical power of the LED using the graphene solution were 25.1, 27.5, 28.2 and 28.8 mW at flow rates of 0, 0.05, 0.2 and 1 mL/min, respectively. The optical power of the LED using the graphene solution was higher than that of the solid conductor. When the flow rate was increased, the optical power of the LED was higher, but the difference was not large.

The optical/electric power transfer rate of the LED was also investigated. The electric currents of the LED using various conductors were measured at 10, 40, 90, 150, and 480 min of LED operation. The supplied electric power was calculated using the following equation:*Pe* = *I* × *V*(4)
where *Pe* is the supplied electric power (mW), *I* is the electric current (mA), and *V* is the voltage (V). The relative optical/electric power transfer rate was calculated using the following equation:*Rt* = *Po*/*Pe*(5)
where *Rt* is the optical/electric power transfer rate, *Po* is the measured optical power, and *Pe* is the supplied electric power. The results are presented in Figure 13. After 10 min of LED operation, the optical/electric power transfer rates were 50.1%, 35.3%, and 34.7% for the solid conductor, the liquid conductor with *d* = 0.5 mm, and the liquid conductor with *d* = 1 mm, respectively. At 60 min, the transfer rates were 45.0%, 42.7%, and 41.9%, respectively. At 120 min, the transfer rates were 40.7%, 46.1%, and 44.8%, respectively. The optical/electric power transfer rate of the liquid conductor was higher than that of the solid conductor after 120 min of LED operation. Therefore, the optical/electric power transfer rate was affected by the temperature of the LED and was improved by the graphene-solution liquid conductor.

## 4. Conclusions

This study proposes an automated graphene-solution circulating system that can efficiently dissipate heat and conduct electricity. The temperature and optical power of an LED were measured over an extended period. The thermal resistance of graphene solution with flow rate was smaller than that of the solid conductor, and the heat transfer of the LED was improved. The difference in temperature of the LED bottom between the LEDs using liquid and solid conductors reached 25 °C. After 120 min of LED operation, the optical power of the LED with the liquid conductor was higher than that of the solid conductor. When the flow rate was increased, the temperature difference of the LED bottom between initial and 480 min was lower, and the optical power of the LED was higher. This result is attributable to the higher temperature of the LED with the solid conductor. Furthermore, after 120 min of LED operation, the optical/electric power transfer rate of the liquid conductor was higher than that of the solid conductor, and the difference between them increased over time. The graphene-solution automated circulating system is a suitable conductor for LEDs and is particularly useful for operation over long periods.

## Figures and Tables

**Figure 1 nanomaterials-11-01719-f001:**
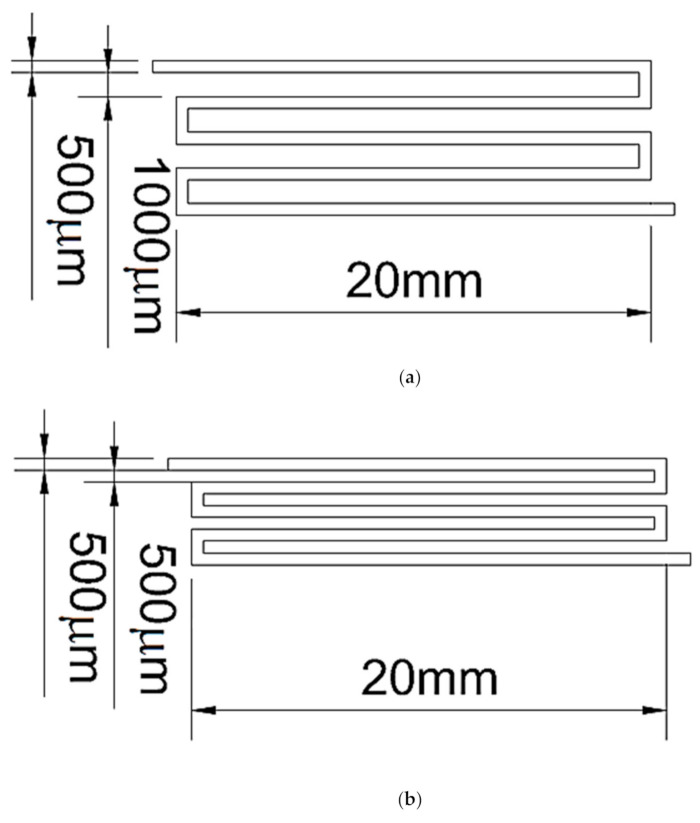
Schematics of the LED chip with a microfluidic channel: (**a**) distance between the channel *d* = 1 mm; (**b**) distance between the channel *d* = 0.5 mm.

**Figure 2 nanomaterials-11-01719-f002:**
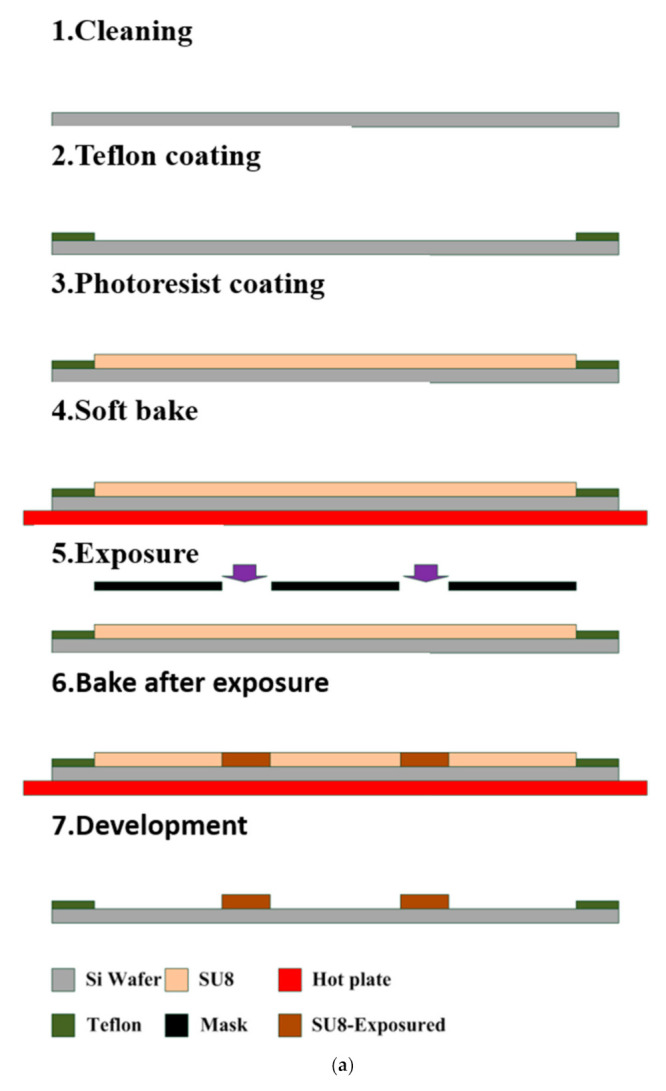
Fabrication of the microfluidic channels: (**a**) SU8 mold chip; (**b**) PDMS substrate microfluidic channel chip.

**Figure 3 nanomaterials-11-01719-f003:**
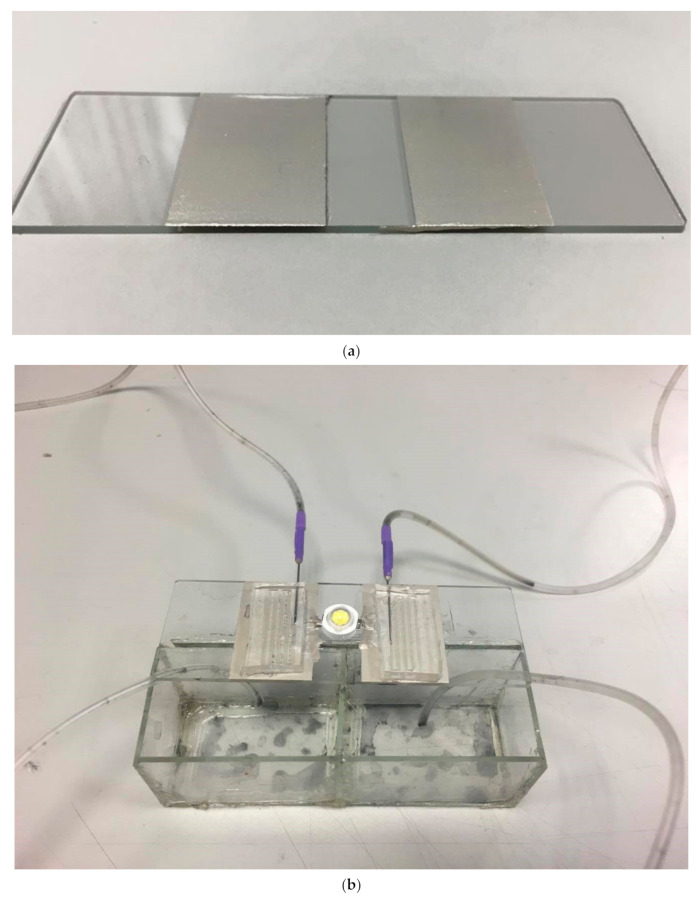
Schematics of the system: (**a**) image of the microscope slide with silver adhesive; (**b**) image of the LED with a liquid circulating system.

**Figure 4 nanomaterials-11-01719-f004:**
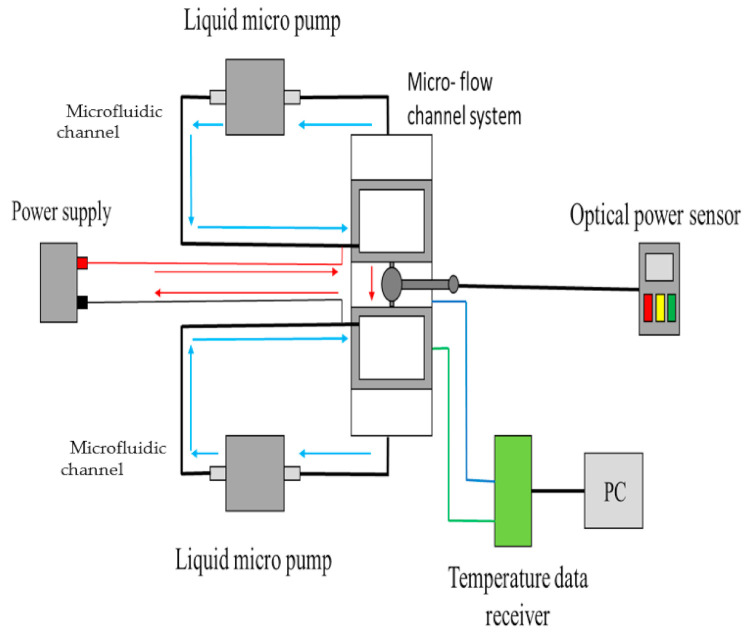
Schematic of the integrated system.

**Figure 5 nanomaterials-11-01719-f005:**
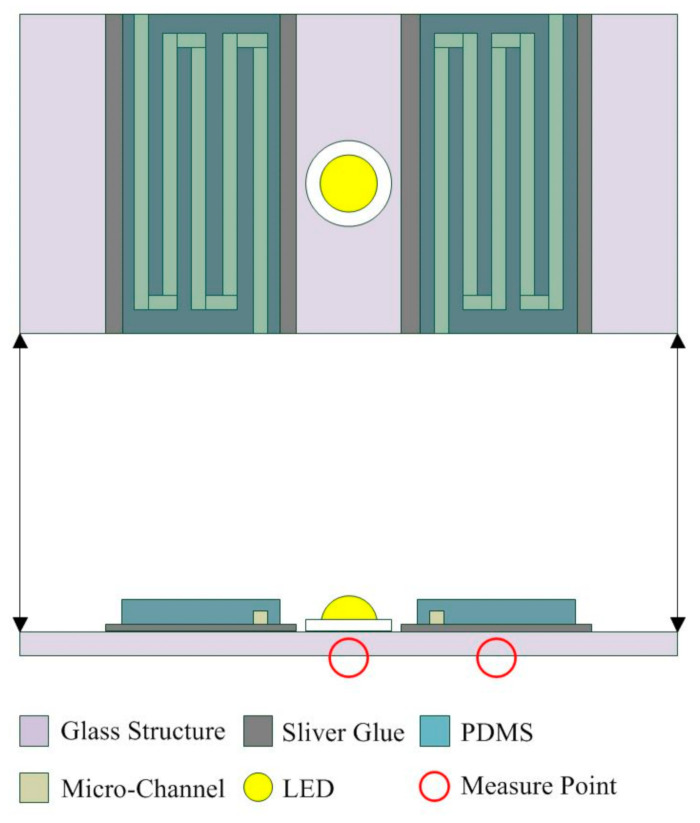
Schematic of the temperature measurement points of the LED bottom and microfluidic channel.

**Figure 6 nanomaterials-11-01719-f006:**
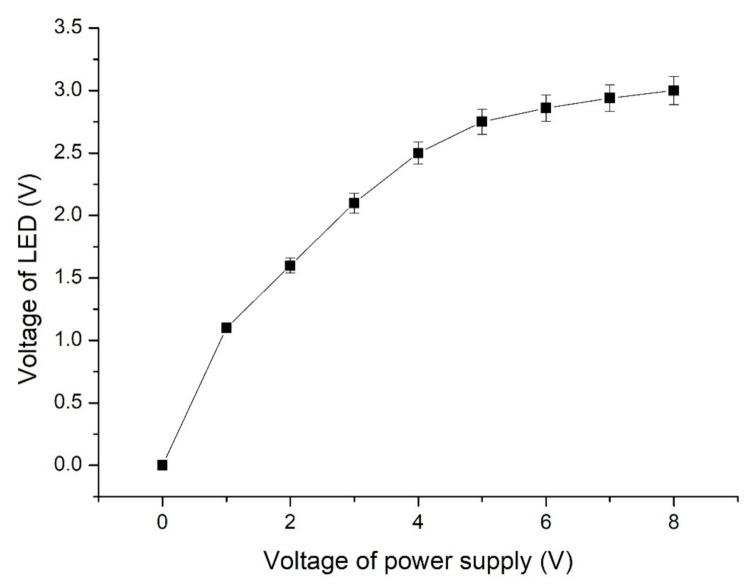
Electric voltages of the LED with the liquid conductor at various power supply electric voltages.

**Figure 7 nanomaterials-11-01719-f007:**
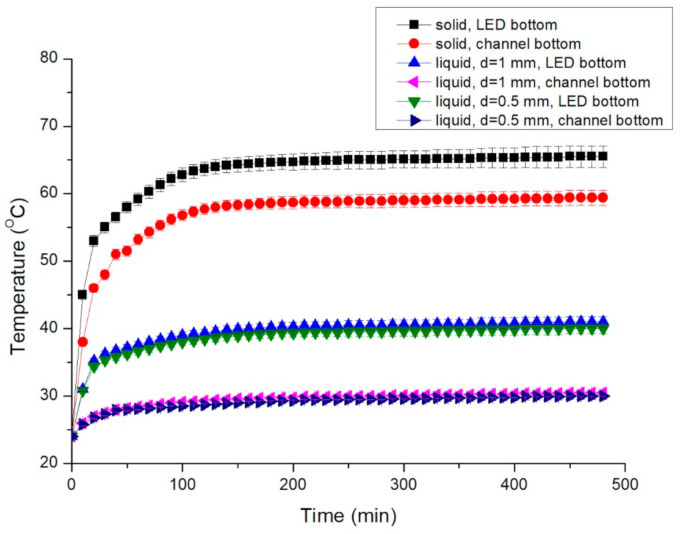
Temperatures of the bottom of the LED bottom and channel bottom with various conductors.

**Figure 8 nanomaterials-11-01719-f008:**
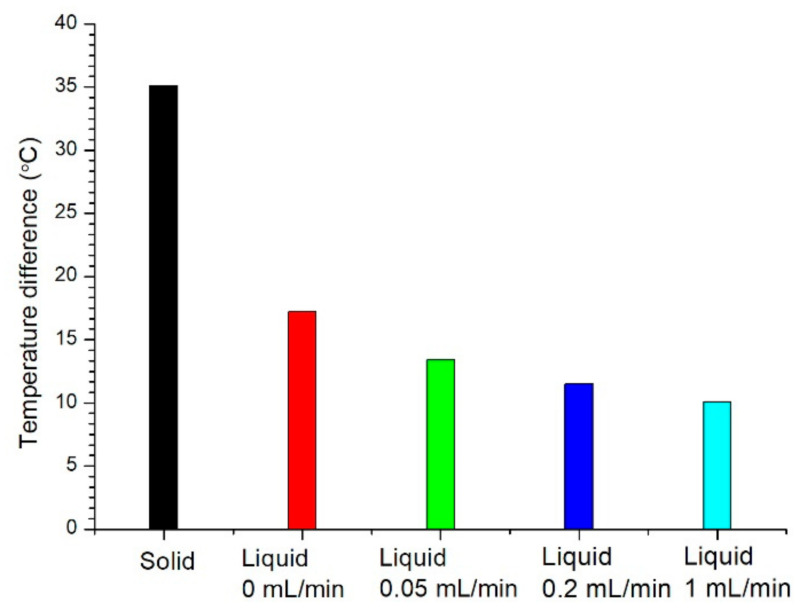
Temperature differences of the LED bottom between initial and 480 min using solid conductor and graphene solution with various flow rates.

**Figure 9 nanomaterials-11-01719-f009:**
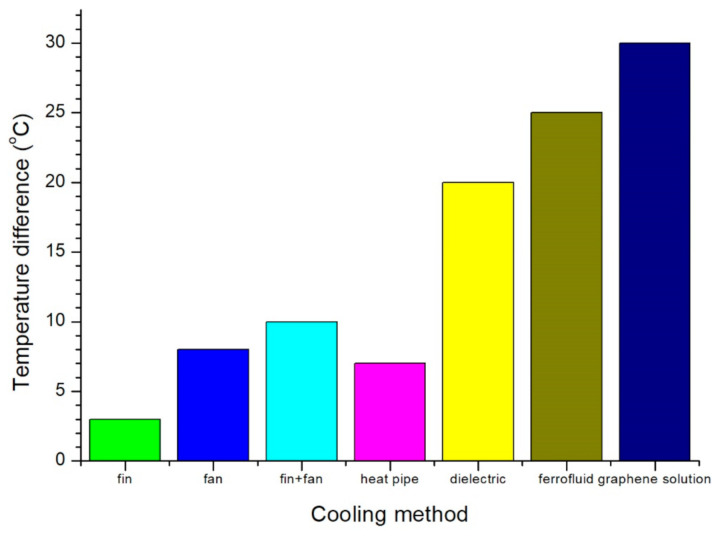
Temperature differences of LED using various cooling methods.

**Figure 10 nanomaterials-11-01719-f010:**
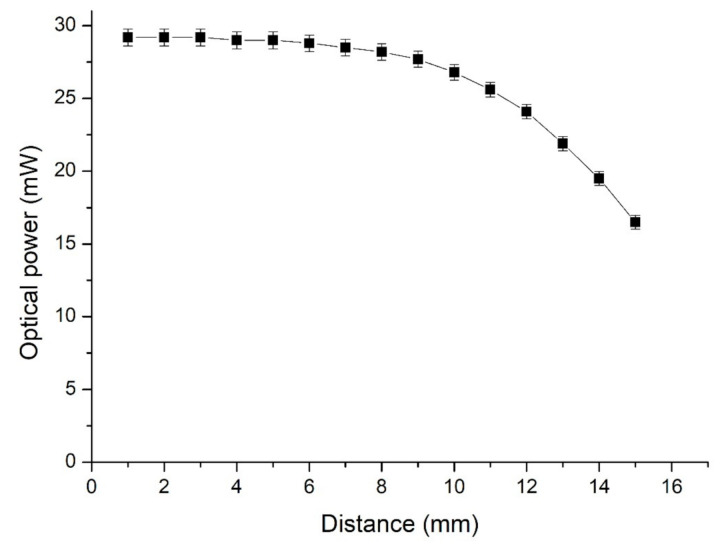
Optical power of the LED chip at various distances between the LED and the power meter.

**Figure 11 nanomaterials-11-01719-f011:**
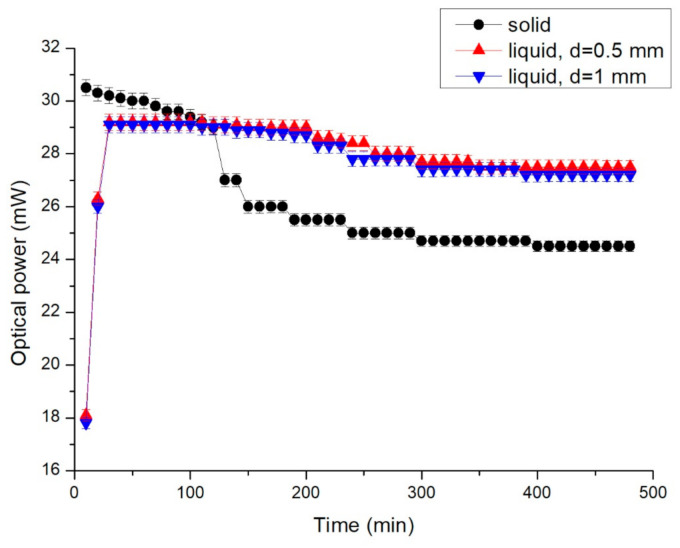
Optical power of the LED chip with various conductors and distances between channels.

**Figure 12 nanomaterials-11-01719-f012:**
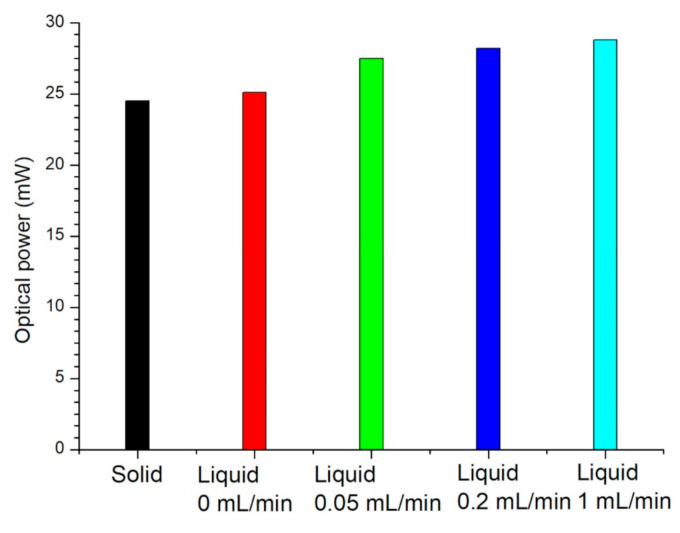
Optical power of the LED at 480 min using solid conductor and graphene solution with various flow rates.

**Figure 13 nanomaterials-11-01719-f013:**
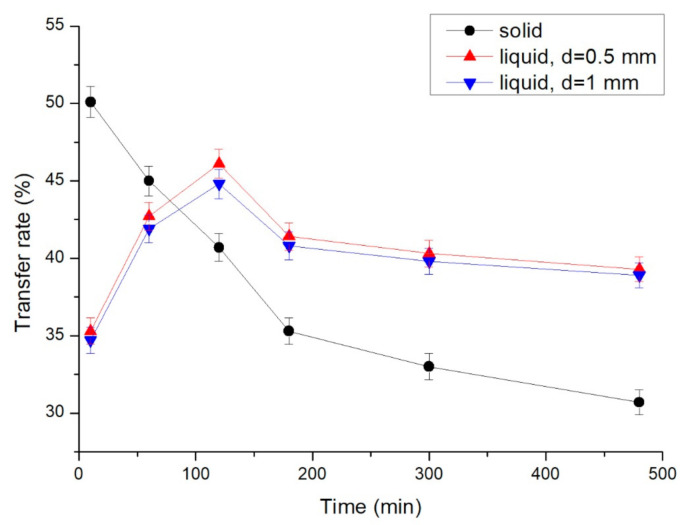
Optical/electric power transfer rate of the LED chip with various conductors and distances between channels.

**Table 1 nanomaterials-11-01719-t001:** Uncertainty analysis of various parameters.

Parameter	Instrument	Uncertainty
Electric resistance and voltage	Digital multimeters	0.1%
Flow rate	Piezoelectric pump	0.5%
Temperature	Resistance temperature detectors	0.1 °C
Optical power	Optical power meter	0.5%

## Data Availability

Authors ensure that data shared are in accordance with consent provided by participants on the use of confidential data. The data presented in this study are available on request from the corresponding author. The data are not publicly available due to the consideration of commercialization.

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
