# Peer review of "Improvement of Temperature and Optical Power of an LED by Using Microfluidic Circulating System of Graphene Solution"

_nanomaterials, 2021, doi:10.3390/nano11071719_

Round 1

Reviewer 1 Report

The authors addressed most of the comments. However, I would like to ask a few more things for clarifying the experimental setup.

  1. The authors responded that the electric voltage of the LED was 2.7 V when the power supply electric voltage was 2.8V, which seems not consistent with Figure 6 and the statement in the manuscript. (“The voltage of the graphene solution-conductor LED was 2.8V when the voltage of the power supply was 8V” - page 8, line #181-183) If Figure 6 and the original statement were correct, there would be significant voltage drop and heat generation in liquid or solid conductor.

  1. The authors said that when the current was fixed at 0.5 A, the voltage of power supply was lower than 1.5V. At this condition, LED will be in off state due to the smaller applied voltage than the turn-on voltage as discussed in the manuscript. This response (0.5A at 1.5V) might conflict with Figure 4 showing that the LED device seems to be connected to the power supply in series with conductor and motors. In the same vein, I could not understand the authors' response “the LED optical power increases by the current when the power supply voltage is fixed at 2.8V”. Electric voltage of the LED is determined by the power supply voltage as seen in Figure 6, and the current through the LED device is determined by the applied voltage to the LED device. Could the authors answer how to control the LED current at a given voltage? Accordingly, the experimental setup and electrical circuit should be more accurately represented in Figure 4. In addition, the authors should give the applied power to the LED device, thus, the power consumption of the LED device.

Author Response

Thanks for the reviewer's comment. The attachment is the response to the reviewer's comment. Thank you very much.

Reviewer 2 Report

This is an interesting study pointing out the efficiency of a graphene solution to cool down a LED with respect to other cooling methods.

However, it would have been useful to explain or propose a scheme for the physical mechanism involved which makes the presence of graphene in solution more efficient for the heat transfer. For example information or demonstration of a change of thermal conductivity versus graphene concentration in the solution would be useful. 

Author Response

(The authors gave the same response as above.)

Round 2

Reviewer 1 Report

The authors' responses can be accepted. 

Author Response

Thanks for the reviewer' comment.

Reviewer 2 Report

I am sorry to be not able to accept this paper. This interesting study of the cooling efficiency using microfluidic circulating system of graphene solution would be more adapted to technical reviews such as Review of Scientific Instruments for example.

Author Response

Dear reviewer 2,

The response to reviewer 2 was included in the revised manuscript.

Thank you very much.

Sincerelt

Yung-Chiang Chung

This manuscript is a resubmission of an earlier submission. The following is a list of the peer review reports and author responses from that submission.

Round 1

Reviewer 1 Report

The authors have studied heat dissipation via graphene solution in microfluidic circulating system for LED. The graphene solution with an automated reclining system has advantages over Ni-based solid conductors in terms of long-term LED operation. However, more detail comparative analysis with reference (solid conductor or water without graphene) is highly suggested for clarifying the strength of the graphene-based liquid conductor.

Comments on below:

  1. Could you provide the electric voltage of the LED with solid conductor depending on the power supply electric voltages for clarity? In addition, where does the voltage drop occur except for LED device? The elements causing the voltage drop will also generate heat so that it may affect the thermal management. In this regard, discussion on the all heat sources in the system may be necessary.
  2. Could you provide more information on solid conductor in terms of geometry (size), volume, and so on? More detail comparisons with solid conductor may be needed to show the effectiveness of liquid conductors.
  3. An LED device is mainly controlled by the current rather than voltage. Therefore, it would be better to compare the optical power of the LED devices with different thermal management methods for a same driven current rather than voltage. In this case, it is also possible to compare the junction temperature, which is a factor that affects the LED operation directly, through the change of the operation voltage. Meanwhile, it would be also good to show how the LED current changes under the same operating voltage condition for inferring the junction temperature change.
  4. Could you comment on the remarkably lower optical power of the LED devices with liquid conductors compared to that with solid conductor for the initial operation region?
  5. The size and layout of the figures are highly necessary to be modified and improved for readability.

Reviewer 2 Report

The authors report the effects of temperature and optical power of an LED integrated onto a microfluidic system using a graphene solution. The authors attempt to demonstrate the characteristics of the dissipation of heat in an LED using the microfluidic system. However, I don't see the novelty of this approach and the system is not systematically characterized, see below comments. For these reasons, I don't recommend accepting this paper to the Journal of Nanomaterials. 

1) In Figures 1, 2, and 3, I don't see any new fabrication methods and tools. 
2) The authors compared temperature variations with microfluidic-based liquid conductions and a solid conductor. It would be more interesting if conducted flow rate vs. temperature and optical power. Also, how does the temperature gradient look like in the channel? Is it possible to show numeral simulation?
3) The authors claim "automated circulating system". The authors used a conventional syringe pump. In what aspect, this can be an automated system. Is this programmable in future work?